# Sparse Attentive Backtracking: Long-Range Credit Assignment in Recurrent Networks

## Abstract

A major drawback of backpropagation through time (BPTT) is the difficulty of learning long-term dependencies, coming from having to propagate credit information backwards through every single step of the forward computation. This makes BPTT both computationally impractical and biologically implausible. For this reason, full backpropagation through time is rarely used on long sequences, and truncated backpropagation through time is used as a heuristic. However, this usually leads to biased estimates of the gradient in which longer term dependencies are ignored. Addressing this issue, we propose an alternative algorithm, Sparse Attentive Backtracking, which might also be related to principles used by brains to learn long-term dependencies. Sparse Attentive Backtracking learns an attention mechanism over the hidden states of the past and selectively backpropagates through paths with high attention weights. This allows the model to learn long term dependencies while only backtracking for a small number of time steps, not just from the recent past but also from attended relevant past states.

## 1 Introduction

Recurrent Neural Networks (RNNs) are state-of-the-art for many machine learning sequence processing tasks. Examples where models based on RNNs shine include speech recognition (Miao et al., 2015; Chan et al., 2016), image captioning (Xu et al., 2015; Vinyals et al., 2015; Lu et al., 2016), machine translation (Bahdanau et al., 2014; Sutskever et al., 2014; Luong et al., 2015), and speech synthesis (Mehri et al., 2016). It is common practice to train these models using backpropagation through time (BPTT), wherein the network states are unrolled in time and gradients are backpropagated through the unrolled graph. Since the parameters of an RNN are shared across the different time steps, BPTT is more prone to vanishing and exploding gradients (Hochreiter, 1991; Bengio et al., 1994; Hochreiter, 1998) than equivalent deep feedforward networks with as many stages. This makes *credit assignment* particularly difficult for events that have occurred many time steps in the past, and thus makes it challenging in practice to capture long-term dependencies in the data (Hochreiter, 1991; Bengio et al., 1994). Having to wait for the end of the sequence in order to compute gradients is neither practical for machines nor animals when the dependencies extend over very long timescales. Training is slowed down considerably by long waiting times, as the rate of convergence crucially depends on how often parameters can be updated.

In practice, proper long-term credit assignment in RNNs is very inconvenient, and it is common practice to employ truncated versions of BPTT for long sequences (Sak et al., 2014; Saon et al., 2014). In truncated BPTT (TBPTT), gradients are backpropagated only for a fixed and limited number of time steps and parameters are updated after each such subsequence. Truncation is often motivated by computational concerns: memory, computation time and the advantage of faster learning obtained when making more frequent updates of the parameters rather than having to wait for the end of the sequence. However, it makes capturing correlations across distant states even harder.

Regular RNNs are parametric: their hidden state vector has a fixed size. We believe that this is a critical element in the classical analysis of the difficulty of learning long-term dependencies (Bengio et al., 1994). Indeed, the fixed state dimension becomes a bottleneck through which information has to flow, both forward and backward.

We thus propose a semi-parametric RNN, where the next state is potentially conditioned on all the previous states of the RNN, making it possible—thanks to attention—to jump through any distance through time. We distinguish three types of states in our proposed semi-parametric RNN:

- The fixed-size *hidden state* $\boldsymbol{h}^{(t)}$, the conventional state of an RNN model at time $t$;

- The monotonically-growing *macrostate* $\mathcal{M} = \{\boldsymbol{m}^{(1)}, \dots, \boldsymbol{m}^{(s)}\}$, the array of all past *microstates*, which plays the role of a random-access memory;

- And the fixed-size *microstate* $\boldsymbol{m}^{(i)}$, which is the $i$th hidden state (one of the $\boldsymbol{h}^{(t)}$) that was chosen for inclusion within the macrostate $\mathcal{M}$.

There are as many hidden states as there are timesteps in the sequence being analyzed by the RNN. A subset of them will become microstates, and this subset is called the macrostate.

The computation of the next hidden state $\boldsymbol{h}^{(t+1)}$ is based on the whole macrostate $\mathcal{M}$, in addition to the external input $\boldsymbol{x}^{(t)}$. The macrostate being variable-length, we must devise a special mechanism to read from this ever-growing array. As a key component of our model, we propose to use an *attention mechanism* over the microstate elements of the macrostate.

The attention mechanism in the above setting may be regarded as providing adaptive, dynamic skip connections: any past *microstate* can be linked, via a dynamic decision, to the current *hidden state*. Skip connections allow information to propagate over very long sequences. Such architectures should naturally make it easier to learn long-term dependencies. We name our algorithm sparse attentive backtracking (SAB). SAB is especially well-suited to sequences in which two parts of a task are closely related yet occur very far apart in time.

Inference in SAB involves examining the macrostate and selecting some of its microstates. Ideally, SAB will not select all microstates, instead attending only to the most salient or relevant ones (e.g., emotionally loaded, in animals). The attention mechanism will select a number of relevant microstates to be incorporated into the hidden state. During training, local backpropagation of gradients happens in a short window of time around the selected microstates only. This allows for the updates to be asynchronous with respect to the time steps we attend to, and credit assignment takes place more globally in the proposed algorithm.

With the proposed framework for SAB, we present the following contributions:

- A principled way of doing sparse credit assignment, based on a semi-parametric RNN.

- A novel way of mitigating exploding and vanishing gradients, based on reducing the number of steps that need to be backtracked through temporal skip connections.

- Competitive results compared to **full** backpropagation through time (BPTT), and much better results as compared to Truncated Backpropagation through time, with significantly shorter truncation windows in our model.

Mechanisms such as SAB may also be biologically plausible. Imagine having taken a wrong turn on a roadtrip and finding out about it several miles later. Our mental focus would most likely shift directly to the location in time and space where we had made the wrong decision, without replaying in reverse the detailed sequence of experienced traffic and landscape impressions. Neurophysiological findings support the existence of such attention mechanisms and their involvement in credit assignment and learning in biological systems. In particular, hippocampal recordings in rats indicate that brief sequences of prior experience are replayed both in the awake resting state and during sleep, both of which conditions are linked to memory consolidation and learning (Foster & Wilson, 2006; Davidson et al., 2009; Gupta et al., 2010). Moreover, it has been observed that these replay events are modulated by the reward an animal does or does not receive at the end of a task in the sense that they are more pronounced in the presence of a reward signal and less pronounced or absent in the absence of a reward signal (Ambrose et al., 2016). Thus, the mental look back into the past seems to occur exactly when credit assignment is to be performed.

## 2 RELATED WORK

### 2.1 TRUNCATED BACKPROPAGATION THROUGH TIME

When training on very long sequences, full backpropagation through time becomes computationally expensive and considerably slows down training by forcing the learner to wait for the end of each (possibly very long sequence) before making a parameter update. A common heuristic is to

backpropagate the loss of a particular time step through only a limited number of time steps, and hence truncate the backpropagation computation graph (Williams & Peng, 1990). While truncated backpropagation through time is heavily used in practice, its inability to perform credit assignment over longer sequences is a limiting factor for this algorithm, resulting in failure cases even in simple tasks, such as the Copying Memory and Adding task in (Hochreiter & Schmidhuber, 1997).

## 2.2 DECOUPLED NEURAL INTERFACES

The Decoupled Neural Interfaces method (Jaderberg et al., 2016) replaces full backpropagation through time with synthetic gradients, which are essentially small networks, mapping the hidden unit values of each layer to an estimator of the gradient of the final loss with respect to that layer. While training the synthetic gradient module requires backpropagation, each layer can make approximate gradient updates for its own parameters in an asynchronous way by using its synthetic gradient module. Thus, the network learns how to do credit assignment for a particular layer from a few examples of the gradients from backpropagation, reducing the total number of times that backpropagation needs to be performed.

## 2.3 APPROXIMATE FORWARD-MODE ONLINE RNNS

Online credit assignment in RNNs without backtracking remains an open research problem. One approach (Ollivier et al., 2015) attempts to solve this problem by estimating gradients using an approximation to forward mode automatic differentiation instead of backpropagation. Forward mode automatic differentiation allows for computing unbiased gradient estimates in an online fashion, however it normally requires storage of the gradient of the current hidden state values with respect to the parameters, which is $O(N^3)$ where $N$ is the number of hidden units. The Unbiased Online Recurrent Optimization (UORO) (Tallec & Ollivier, 2017) method gets around this by updating a rank-1 approximation to this gradient tensor, which is shown to keep the estimate of the gradient unbiased, but potentially at the risk of increasing the variance of the gradient estimator.

## 2.4 SKIP-CONNECTIONS AND GRADIENT FLOW

Neural architectures such as Residual Networks (He et al., 2016) and Dense Networks (Huang et al., 2016) allow information to skip over convolutional processing blocks of an underlying convolutional network architecture. In the case of Residual Networks identity connections are used to skip over convolutional processing blocks and this information is recombined using addition. This construction provably mitigates the vanishing gradient problem by allowing the gradient at any given layer to be bounded. Densely-connected convolutional networks alleviate the vanishing gradient problem by allowing a direct path from any point in the network to the output. In contrast, here we propose and explore what one might regard as a form of dynamic skip connection, modulated by an attention mechanism.

## 3 SPARSE ATTENTIVE BACKTRACKING

We now introduce the idea of Sparse Attentive Backtracking (SAB). Classical RNN models such as those based on LSTMs or GRUs only use the previous hidden state in the computation of the next one, and therefore struggle with extremely long-range dependencies. SAB sidesteps this limitation by additionally allowing the model to select and use (a subset of) *any* of the past microstates in the computation of the next hidden state. In doing so the model may potentially reference microstates computed arbitrarily long ago in time.

Since the classic RNN models do not support such operations on their past, we make a few architectural additions. On the forward pass of a training step, a mechanism is introduced that selects microstates from the macrostate, summarizes them, then incorporates this summary into the next hidden state. The hidden state may or may not become a microstate. On the backward pass, the gradient is allowed to flow not only through the (truncated) master chain linking consecutive hidden states, but also to the microstates which are selected in the forward pass.

In the forward pass, the microstate selection process can be denser or sparser, and the summarization and incorporation can be more or less sophisticated. In the backward pass, the gating of gradient flow from a hidden state to its ancestor microstates can also be denser or sparser, although it can be no denser than the forward pass was.

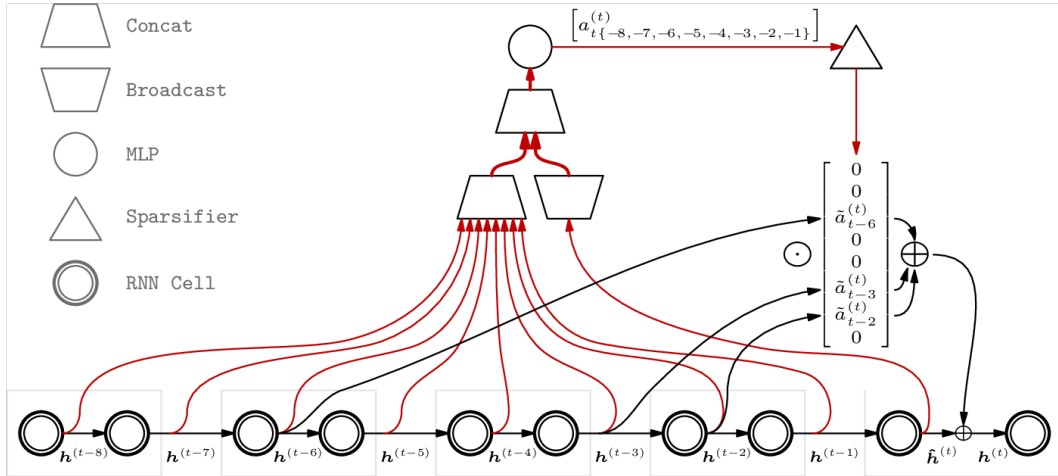

Figure 1: This figure illustrates the forward pass in SAB for the configuration $k_{top} = 3$, $k_{att} = 1$, $k_{trunc} = 2$. This involves Microstate Selection (§ 3.2), Summarization of microstates (§ 3.3), and incorporation into next microstate (§ 3.4). Red arrows depict how attention weights $\left[ a_{t\{-8,-7,-6,-5,-4,-3,-2,-1\}}^{(t)} \right]$ are evaluated, first by broadcasting the current provisional hidden state $\hat{h}^{(t)}$ against the macrostate (which, in the presented case of $k_{att} = 1$, consists of all past hidden states), concatenating, then passing the result to an MLP. The attention weights are then run through the sparsifier which selects the $k_{top} = 3$ attention weights, while the others are zeroed out. Black arrows show the microstates corresponding to the non-zero sparse attention weights $\{\tilde{a}_{t-6}^{(t)}, \tilde{a}_{t-3}^{(t)}, \tilde{a}_{t-2}^{(t)}\}$, namely $\{m^{(t-6)} = h^{(t-6)}, m^{(t-3)} = h^{(t-3)}, m^{(t-2)} = h^{(t-2)}\}$, being weighted, summed, then incorporated into $\hat{h}^{(t)}$ to compute the current final hidden state $h^{(t)}$.

For instance, it is possible for the forward pass to be dense, incorporating a summary of all microstates, but for the backward pass to be sparse, only allowing gradient flow to some of the microstate contributors to the hidden state (*Dense Forward, Sparse Backward*). Another possibility is for the forward pass to be sparse, making only a few, hard, microstate selections for the summary. In this case, the backward pass will necessarily also be sparse, since few microstates will have contributed to the hidden state, and therefore to the loss (*Sparse Forward, Sparse Backward*).

Noteworthy is that not *all* hidden states need be eligible to become microstates. In practice, we have found that restricting the pool of eligible hidden states to only every $k_{att}$'th one still works well, while reducing both memory and computation expense. Such an increase in the granularity of microstate selection can also improve performance, by preventing the model from attending exclusively to the most recent hidden states and temporally spreading microstates out from each other.

### 3.1 UNDERLYING RNN ARCHITECTURE

The SAB algorithm is widely applicable, and is compatible with numerous RNN architectures, including vanilla, GRU and LSTM models. However, since it necessarily requires altering the hidden-to-hidden transition function substantially, it's currently incompatible with the accelerated RNN kernels offered by e.g. NVIDIA on its GPU devices through cuDNN library (Chetlur et al., 2014).

For vanilla and GRU-inspired RNN architectures, SAB's selection and incorporation mechanisms operate over the (hidden) state. For the LSTM architecture, which we adopt for our experiments, they operate over the hidden state but *not* the cell state.

### 3.2 MICROSTATE SELECTION

The microstate selection mechanism determines which microstate subset of the macrostate will be selected for summarization on the forward pass of the RNN, and which subset of that subset will receive gradient on the backward pass during training. This makes it the core of the attention mechanism of a SAB implementation.

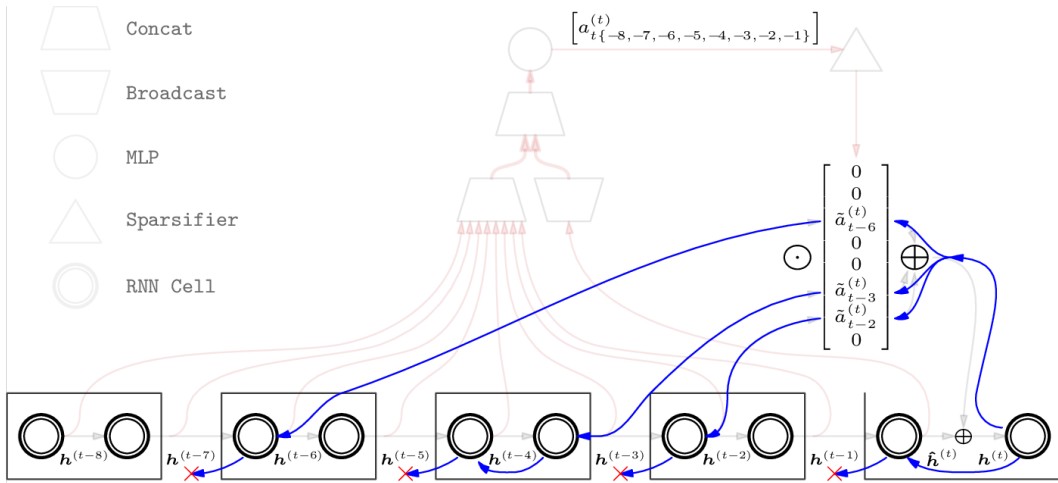

Figure 2: This figure illustrates the backward pass in SAB for the configuration $k_{top} = 3$, $k_{att} = 1$, $k_{trunc} = 2$. The gradients are passed to the microstates selected in the forward pass and a local truncated backprop is performed around those microstates. Blue arrows show the gradient flow in the backward pass. Red crosses indicate TBPTT truncation points, where the gradient stops being backpropagated.

While the selection mechanism may use hard-coded attention heuristics, there is no reason why the microstate selection mechanism could not itself be a (deep) neural network trained alongside the RNN model over which it operates.

In the models we use here, the selection mechanism is chosen to be a 1-hidden-layer Linear-Tanh-Linear MLP that computes a scalar *attention weight* $a_i$ for each eligible microstate vector $\boldsymbol{m}^{(i)}$, and a sparsifier that masks out all but the $k_{top}$ greatest attention weights, producing the *sparse attention weights* $\tilde{a}_i$. We empirically demonstrate that even this simple mechanism learns to focus on past time steps relevant to the current one, thus successfully performing credit assignment. The use of a higher complexity model here would be an interesting avenue for future research.

### 3.3 SUMMARIZATION OF MICROSTATES

The selected microstates must be somehow combined into a fixed-size summary for incorporation into the next hidden state. While many options exist for doing so, we choose to simply perform a summation of the microstates, weighted by their sparsified attention weight $\tilde{a}_i$.

### 3.4 INCORPORATION INTO NEXT MICROSTATE

Lastly, the summary must be incorporated into the hidden state. Again, multiple options exist, such as addition (as done in ResNets) or concatenation (as done in DenseNets).

For our purposes we choose to simply sum the summary into the *provisional* hidden state output $\hat{\boldsymbol{h}}^{(t)}$ computed by the LSTM cell to produce the *final* hidden state $\boldsymbol{h}^{(t)}$ that will be conditioned upon in the next timestep.

### 3.5 IMPLEMENTATION DETAILS

We now give the equations for the specific SAB-augmented LSTM model we use in our experiments.

At time $t$, the underlying LSTM receives a vector of hidden states $\boldsymbol{h}^{(t-1)}$, a vector of cell states $\boldsymbol{c}^{(t-1)}$ and an input $\boldsymbol{x}^{(t)}$, and computes a *provisional* hidden state vector $\hat{\boldsymbol{h}}^{(t)}$ that also serves as a provisional output.

We next use an attention mechanism that is similar to (Bahdanau et al., 2014), but modified to produce sparse discrete attention decisions. First, the provisional hidden state vector $\hat{\boldsymbol{h}}^{(t)}$ is concatenated to each microstate vector $\boldsymbol{m}^{(i)}$. Then, an MLP maps each such concatenated vector to an attention weight $a_i^{(t)}$ representing the salience of the microstate $i$ at the current time $t$. This can be

expressed as:

$$a_{i_1}^{(t)} = \boldsymbol{w}_1^\top \boldsymbol{m}^{(i)} + \boldsymbol{w}_2^\top \hat{\boldsymbol{h}}^{(t)} \tag{1}$$

$$a_i^{(t)} = \boldsymbol{w}_3^\top \tanh(\boldsymbol{a}_{i_1}^{(t)}) \tag{2}$$

where the weights matrices $\boldsymbol{w}_1$, $\boldsymbol{w}_2$ and $\boldsymbol{w}_3$ are learned parameters.

Following this, we apply a piece-wise linear function that sparsifies the attention while making discrete decisions. (This is different from typical attention mechanisms that normalize attention weights using a Softmax function (Bahdanau et al., 2014), whose output is never sparse). Let $a_{k\text{top}}^{(t)}$ be the $k_{\text{top}}$th greatest-valued attention weight at time $t$; then the sparsified attention weights are computed as

$$\tilde{a}_i^{(t)} = \text{ReLU}\left(a_i^{(t)} - a_{k\text{top}}^{(t)}\right) \tag{3}$$

This has the effect of zeroing all attention weights less than $a_{k\text{top}}^{(t)}$, thus masking out all but the $k_{\text{top}}$ most salient microstates in $\mathcal{M}$. The few selected microstates receive gradient information, while no gradient flows to the rest.

A summary vector $\boldsymbol{s}^{(t)}$ is then obtained using a weighted sum over the macrostate, employing the sparse attention weights:

$$\boldsymbol{s}^{(t)} = \sum_{\boldsymbol{m}^{(i)} \in \mathcal{M}} \tilde{a}_i^{(t)} \boldsymbol{m}^{(i)} \tag{4}$$

Given that this sum is very sparse, the summary operation is very fast.

To incorporate the summary into the *final* hidden state at timestep $t$, we simply sum the summary and the *provisional* hidden state:

$$\boldsymbol{h}^{(t)} = \hat{\boldsymbol{h}}^{(t)} + \boldsymbol{s}^{(t)} \tag{5}$$

Lastly, to compute the output at the time step $t$, we concatenate $\boldsymbol{h}^{(t)}$ and the sparse attention weights $\tilde{\boldsymbol{a}}^{(t)}$, then apply an affine output transform to compute the output. This can be equivalently expressed as:

$$\boldsymbol{y}^{(t)} = \boldsymbol{V}_1^\top \boldsymbol{h}^{(t)} + \boldsymbol{V}_2^\top \tilde{\boldsymbol{a}}^{(t)} + \boldsymbol{b} \tag{6}$$

where the weights matrices $\boldsymbol{V}_1$ and $\boldsymbol{V}_2$ and bias vector $\boldsymbol{b}$ are learned parameters.

In summary, for a given time step $t$, a hidden state $\boldsymbol{h}^{(i)}$ selected by the hard-attention mechanism has two paths contributing to the hidden states $\boldsymbol{h}^{(t)}$ in the forward pass. One path is the regular sequential forward path in an RNN; the other path is through the dynamic skip connections in the attention mechanism. When we perform backpropagation through the skip connections, gradient only flows from $\boldsymbol{h}^{(t)}$ to microstates $\boldsymbol{m}^{(i)}$ selected by the attention mechanism (those for which $\tilde{a}_i^{(t)} > 0$).

### 3.5.1 Attention Mechanism Notes

In the preparation of this work, it was discovered that the attention mechanism absolutely *must* include a non-linearity in the computation of the raw attention weights $a_i^{(t)}$. Our failure to do so in an early iteration of the work resulted in a catastrophic cancellation in the subsequent sparsification of the weights to $\tilde{a}_i^{(t)}$. This is because in (5), a rectified difference between $a_i^{(t)}$ is computed to zero all but the $k_{top}$ greatest attention weights. Subtraction is linear; And since our earlier attention mechanism was linear as well, it could be separated into two parts, a first half to which only the microstate $\boldsymbol{m}^{(i)}$ contributed and a second half to which only the hidden state $\boldsymbol{h}^{(t)}$ contributed. This second half of the contribution is catastrophically cancelled in the difference $a_i^{(t)} - a_{k\text{top}}^{(t)}$, because it was computed from the same $\boldsymbol{h}^{(t)}$ for both, and therefore equal.

## 4 Experiments

We now report and discuss the results of an empirical study that analyses the performance of SAB using five different tasks. We first study synthetic tasks—the copying and adding problems (Hochreiter & Schmidhuber, 1997) designed to measure models' abilities to learn long-term dependencies—meant to confirm that SAB can successfully perform credit assignment for events that have occurred many time steps in the past. We then study more realistic tasks and larger datasets.

**Baselines** We compare the quantitative performance of our model against two LSTM baselines (Hochreiter & Schmidhuber, 1997). The first is trained with backpropagation through time (BPTT) and the second is trained using truncated backpropagation through time (TBTPP). Both methods are trained using teacher forcing (Williams & Zipser, 1989). We also used gradient clipping (that is, we clip the gradients to 1 to avoid exploding gradients). Hyperparameters that are task-specific are discussed in the tasks' respective subsections, other hyperparameters that are also used by SAB and that we set to the same value are discussed below.
Compared to standard RNNs, our model has two additional hyperparameters:

- $k_{top}$, the number of most-salient microstates to select at each time step for passing gradients in the backward pass

- $k_{att}$, the granularity of attention. Every $k_{att}$th hidden state is chosen to be a microstate. The special case $k_{att} = 1$ corresponds to choosing all hidden states to be microstates as well.

In addition, we also study the impact of the TBPTT truncation length, which we denote as $k_{trunc}$. This determines how many timesteps backwards to propagate gradients through in the backward pass. This effect of this hyperparameter will also be studies for the LSTM with TBTPP baseline.

For all experiments we used a learning rate of 0.001 with the Adam (Kingma & Ba, 2014) optimizer unless otherwise stated. For SAB, we attend to every second hidden states, i.e. $k_{att}$=2, unless otherwise stated.

Our main findings are:

1. SAB performs almost optimally and significantly outperforms both full backpropagtion through time (BPTT), and truncated backpropagation through time (TBPTT) on the synthetic copying task.

2. For the synthetic adding, two language modelling task (using PennTree Bank and Text8), and permuted sequential MNIST classification tasks, SAB reaches the performance of BPTT and outperforms TBPTT. In addition, for the adding task, SAB outperforms TBPTT using much shorter truncation lengths.

## 4.1 THE COPYING MEMORY PROBLEM

The copying memory task tests the model's ability to memorize salient information for long time periods. We follow the setup of the copying memory problem from Hochreiter & Schmidhuber (1997). In details, the network is given a sequence of $T + 20$ inputs consisting of: a) 10 (randomly generated) digits (digits 1 to 8) followed by; b) $T$ blank inputs followed by; c) a special end-of-sequence character followed by; d) 10 additional blank inputs. After the end-of-sequence character the network must output a copy of the initial 10 digits.

Tables 1, 2, and 3 report both accuracy and cross-entropy (CE) of the models' predictions on unseen sequences. We note that SAB is able to learn this copy task almost perfectly for all sequence-lengths $T$. Further, SAB outperforms all baselines. This is particularly noticeable for longer sequences, for example, when $T$ is 300 the best baseline achieves 35.9% accuracy versus SAB's 98.9%.

To better understand the learning process of SAB, we visualized the attention weights while learning the copying task ($T = 200$, $k_{trunc} = 10$, $k_{top} = 10$). Figure 3 (appendix) shows the attention weights (averaged over a single mini-batch) at three different learning stages of training, all within the first epoch. We note that the attention quickly (and correctly) focuses on the first ten timesteps which contain the input digits. Furthermore, we experimented with LSTM with self-attention trained using full BPTT. The setup is very similar to unidirectional LSTM with self-attention Lin et al. (2017). Due to GPU memory constraints, we are only able to run this experiment up to small sequence lengths. For T=200 and T = 300, we could see that SAB performs comparably to LSTM with full self-attention trained with full BPTT.

## 4.2 THE ADDING TASK

The adding task requires the model to sum two specific entries in a sequence of $T$ (input) entries (Hochreiter & Schmidhuber, 1997). In the spirit of the copying task, larger values of $T$ will require the model to keep track of longer-term dependencies. The exact setup is as follows. Each example in the task consists of 2 input vectors of length $T$. The first, is a vector of uniformly generated values between 0 and 1. The second vector encodes binary a mask that indicates which 2 entries in the

| Method | Copy Length (T) | Accuracy (%) | CE (last 10 chars) | Cross Entropy |
|---|---|---|---|---|
| LSTM (full BPTT) | 100 | 98.8 | 0.030 | 0.002 |
| LSTM (TBPTT, $k_{trunc}$= 5) | 100 | 31.0 | 1.737 | 0.145 |
| LSTM (TBPTT, $k_{trunc}$ = 10) | 100 | 29.6 | 1.772 | 0.148 |
| LSTM (TBPTT, $k_{trunc}$ = 20) | 100 | 30.5 | 1.714 | 0.143 |
| SAB ($k_{trunc}$=1, $k_{top}$=1) | 100 | 57.9 | 1.041 | 0.087 |
| SAB ($k_{trunc}$=1, $k_{top}$=5) | 100 | **100.0** | **0.001** | **0.000** |
| SAB ($k_{trunc}$=5, $k_{top}$=5) | 100 | **100.0** | **0.000** | **0.000** |
| SAB ($k_{trunc}$=10, $k_{top}$=10) | 100 | **100.0** | **0.000** | **0.001** |

Table 1: Test accuracy and cross-entropy loss performance on the **copying task** with sequence lengths of $T = 100$. Models that use TBPTT cannot solve this task while SAB and BPTT can both achieve optimal performance.

| Method | Copy Length (T) | Accuracy (%) | CE (last 10 chars) | Cross Entropy |
|---|---|---|---|---|
| LSTM (full BPTT) | 200 | 56.0 | 1.07 | 0.046 |
| LSTM (TBPTT, $k_{trunc}$= 5) | 200 | 17.1 | 2.03 | 0.092 |
| LSTM (TBPTT, $k_{trunc}$= 10) | 200 | 20.2 | 1.98 | 0.090 |
| LSTM (TBPTT, $k_{trunc}$= 20) | 200 | 35.8 | 1.61 | 0.073 |
| LSTM(TBPTT, $k_{trunc}$=150) | 200 | 35.0 | 1.596 | 0.073 |
| LSTM (full BPTT) + full self-attention | 200 | 100.0 | 0.001 | 0.000 |
| SAB ($k_{trunc}$=1, $k_{top}$=1) | 200 | 39.9 | 1.516 | 0.069 |
| SAB ($k_{trunc}$=5, $k_{top}$=5) | 200 | **100.0** | **0.000** | **0.000** |
| SAB ($k_{trunc}$=10, $k_{top}$=10) | 200 | **100.0** | **0.000** | **0.000** |

Table 2: Test accuracy and cross-entropy loss performance on the **copying task** with sequence lengths of $T = 200$. Different configurations of SAB all reach near optimal performance.

| Method | Copy Length (T) | Accuracy (%) | CE (last 10 chars) | Cross Entropy |
|---|---|---|---|---|
| LSTM (full BPTT) | 300 | 35.9 | 0.197 | 0.047 |
| LSTM (TBTT, $k_{trunc}$= 1) | 300 | 14.0 | 2.077 | 0.065 |
| LSTM (TBPTT, $k_{trunc}$= 20) | 300 | 25.7 | 1.848 | 0.197 |
| LSTM (TBPTT, $k_{trunc}$= 150) | 300 | 24.4 | 1.857 | 0.058 |
| LSTM (full BPTT) + full self-attention | 300 | 100.0 | 0.001 | 0.000 |
| SAB ($k_{trunc}$=1, $k_{top}$=1) | 300 | 43.1 | 0.231 | 0.045 |
| SAB ($k_{trunc}$=1, $k_{top}$=5) | 300 | 89.1 | 0.383 | 0.012 |
| SAB ($k_{trunc}$=5, $k_{top}$=5) | 300 | **99.9** | **0.007** | **0.001** |

Table 3: Test accuracy and cross-entropy loss performance on **copying task** with sequence lengths of $T = 300$. On these long sequences SAB's performance can still be very close to optimal.

first input to sum (it consists of $T - 2$ zeros and 2 ones). The mask is randomly generated with the constraint that masked-in entries must be from different halves of the first input vector.

Tables 4 and 5 report the cross-entropy (CE) of the model's predictions on unseen sequences. When $T = 200$, SAB's performance is similar to the best performance of both baselines. With even longer sequences ($T = 400$), SAB outperforms the TBPTT but is outperformed by BPTT.

### 4.3   CHARACTER LEVEL PENN TREEBANK (PTB)

We evaluate our model on language modelling task using the Penn TreeBank dataset (Marcus et al., 1993). Our LSTM baselines use 1000 hidden units and a learning rate of 0.002. We used non-overlapping sequences of 100 in the batches of 32. We trained SAB for 100 epochs.

We evaluate the performance of our model using the bits-per-character (BPC) metric. As shown in Table 6, we perform slightly worse than BPTT, but better than TBPTT.

| Method | Adding Length (T) | Cross Entropy |
|---|---|---|
| LSTM (full BPTT) | 200 | 0.0000 |
| LSTM (TBPTT, $k_{trunc}$= 20) | 200 | 0.0011 |
| LSTM (TBPTT, $k_{trunc}$=50) | 200 | 0.0003 |
| SAB ($k_{trunc}$=5, $k_{top}$=5) | 200 | **0.0000** |
| SAB ($k_{trunc}$=10, $k_{top}$=10) | 200 | **0.0000** |

Table 4: Performance on unseen sequences of the $T = 200$ **adding task**. We note that all methods have configurations that allow them to perform near optimally.

| Method | Adding Length (T) | Cross Entropy |
|---|---|---|
| LSTM (full BPTT) | 400 | 0.00000 |
| LSTM (TBPTT, $k_{trunc}$=100) | 400 | 0.00068 |
| SAB ($k_{trunc}$=5, $k_{top}$=10, $k_{att}$=5) | 400 | 0.00023 |
| SAB ($k_{trunc}$=10, $k_{top}$=10, $k_{att}$=5) | 400 | **0.00001** |

Table 5: Performance on unseen sequences of the $T = 400$ **adding task**. BPTT slightly outperforms SAB which outperforms TBPTT.

| Method | Valid BPC | Test BPC |
|---|---|---|
| LSTM (full BPTT) | 1.47 | 1.38 |
| LSTM (TBPTT, $k_{trunc}$=1) | 1.57 | 1.47 |
| LSTM (TBPTT, $k_{trunc}$=5) | 1.54 | 1.44 |
| LSTM (TBPTT, $k_{trunc}$=20) | 1.52 | 1.43 |
| SAB (TBPTT, $k_{trunc}$=10, $k_{top}$ =10, $k_{att}$ = 10) | 1.487 | 1.402 |
| SAB (TBPTT, $k_{trunc}$=20, $k_{top}$ =5, $k_{att}$ = 20) | 1.484 | 1.394 |
| SAB (TBPTT, $k_{trunc}$=20, $k_{top}$ =10, $k_{att}$ = 20) | 1.480 | 1.390 |

Table 6: BPC evaluation on the validation set of the character-level PTB (lower is better).

## 4.4 TEXT8

This dataset is derived from the text of Wikipedia and consists of a sequence of a total of 100M characters (non-alphabetical and non-space characters were removed). We follow the setup of Mikolov et al. (2012); use the first 90M characters for training, the next 5M for validation and the final 5M characters for testing. We train on non-overlapping sequences of length 180. Due to computational constraints, all baselines use 1000 hidden units. We trained all models using a batch size of 64. We trained SAB for a maximum of 30 epochs. We have not done any hyperparameter search for our model as it's computationally expensive.

Table 7 reports BPC of the model's predictions on the validation and test sets. Note that SAB's performance closely matches BPTT, and also significantly outperforms TBPTT.

| Method | Valid BPC | Test BPC |
|---|---|---|
| LSTM (full BPTT) | 1.54 | 1.51 |
| LSTM (TBPTT, $k_{trunc}$=5) | 1.64 | 1.60 |
| SAB ($k_{trunc}$=5, $k_{top}$=5, $k_{att}$=5) | 1.56 | 1.53 |

Table 7: Bit-per-character (BPC) Results on the validation and test set for Text8 (lower is better).

## 4.5 PERMUTED PIXEL-BY-PIXEL MNIST

Our last task is a sequential version of the MNIST classification dataset. The task involves predicting the label of the image after being given the image pixel by pixel (pixels are processed in a fixed random order.). All models use an LSTM with 128 hidden units. The prediction is produced by

passing the final hidden state of the network into a softmax. We used a learning rate of 0.001. We trained our model for about 100 epochs, and did early stopping based on the validation set. Table 8 shows that SAB performs about as well as BPTT.

| Method | Valid Accuracy (%) | Test Accuracy (%) |
|---|---|---|
| LSTM (full BPTT) | 91.2 | 90.3 |
| SAB ($k_{trunc}$=14, $k_{top}$=5, $k_{att}$ = 14) | 90.6 | 89.8 |
| SAB ($k_{trunc}$=14, $k_{top}$=10, $k_{att}$ = 14) | 92.2 | 90.9 |
| SAB ($k_{trunc}$=10, $k_{top}$=10, $k_{att}$ = 10) | 92.2 | **91.1** |

Table 8: Test and validation accuracy for the sequential MNIST classification task. The performance of all methods is similar on this task.

## 5 FUTURE WORK

An interesting direction for future development of the Sparse Attentive Backtracking method from the machine learning standpoint would be improving the computational efficiency when the sequences in question are very long. Since the Sparse Attentive Backtracking method uses self-attention on every step, the memory requirement grows linearly in the length of the sequence and computing the attention mechanism requires computing a scalar between the current hidden states and all previous hidden states (to determine where to attend). It might be possible to reduce the memory requirement by using a hierarchical model as done by Chandar et al. (2016), and then recomputing the states for the lower levels of the hierarchy only when our attention mechanism looks at the corresponding higher level of the hierarchy. It might also be possible to reduce the computational cost of the attention mechanism by considering a maximum inner product search algorithm (Shrivastava & Li, 2014), instead of naively computing the inner product with all hidden states values in the past.

## 6 CONCLUSION

Improving the modeling of long-term dependencies is a central challenge in sequence modeling, and the exact gradient computation by BPTT is not biologically plausible as well as inconvenient computationally for realistic applications. Because of this, the most widely used algorithm for training recurrent neural networks on long sequences is truncated backpropagation through time, which is known to produced biased estimates of the gradient (Tallec & Ollivier, 2017), focusing on short-term dependencies. We have proposed Sparse Attentive Backtracking, a new biologically motivated algorithm which aims to combine the strengths of full backpropagation through time and truncated backpropagation through time. It does so by only backpropagating gradients through paths selected by its attention mechanism. This allows the RNN to learn long-term dependencies, as with full backpropagation through time, while still allowing it to only backtrack for a few steps, as with truncated backpropagation through time, thus making it possible to update weights as frequently as needed rather than having to wait for the end of very long sequences.

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

## 7 APPENDIX

### 7.1 COMPUTATIONAL COMPLEXITY OF *SAB*

The time complexity of the forward pass of both training and inference in SAB is $O(tn^2)$, with $t$ the number of timesteps and $n$ the size of the hidden state, although our current implementation scales as $O(t^2n^2)$. The space complexity of the forward pass of training is unchanged at $O(tn)$, but the space complexity of inference in SAB is now $O(tn)$ rather than $O(n)$. However, the time cost of the backward pass of training cost is very difficult to formulate. Hidden states depend on a sparse subset of past microstates, but each of those past microstates may itself depend on several other, even earlier microstates. The web of active connections is, therefore, akin to a directed acyclic graph, and it is quite possible in the worst case for a backpropagation starting at the last hidden state to touch all past microstates several times. However, if the number of microstates truly relevant to a task is low, the attention mechanism will repeatedly focus on them to the exclusion of all others, and pathological runtimes will not be encountered.

### 7.2 GRADIENT FLOW

Our method approximates the true gradient but in a sense it's no different than the kind of approximation made with truncated gradient, except that instead of truncating to the last $k_{trunc}$ time steps, we truncate to one skip-step in the past, which can be arbitrarily far in the past. This provides a way of combating exploding and vanishing gradient problems by learning long-term dependencies. To verify the fact, we ran our model on all the datasets (Text8, Pixel-By-Pixel MNIST, char level PTB) with and without gradient clipping. We empirically found, that we need to use gradient clipping only for text8 dataset, for all the other datasets we observed little or no difference with gradient clipping.

### 7.3 ATTENTION WEIGHT PLOTS

We visualize how the attention weights changes during training for the Copying Memory Task in section 4.1. The attention weights are averaged over the batch. The salient information in a copying task are in the first 10 steps. The figure shows how the attention learns to move towards and concentrate on the beginning of the sequence as training procedes. Note these all happened with the first epoch of training, such that the model learns in a reasonable amount of time.

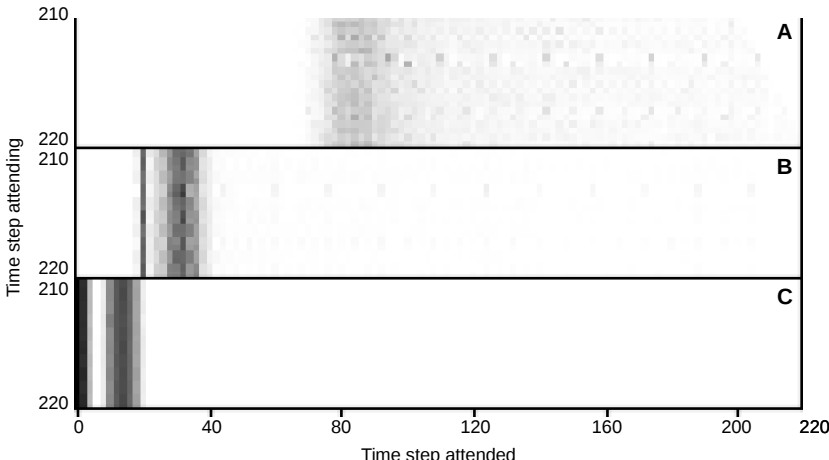

Figure 3: This figure shows how attention weights change over time for the Copying Task of copy length 200. The vertical axis is the time step attending from timestep 210 to timestep 220. The horizontal axis is the time step being attended. The top most subfigure A is the attention plot for iteration 400 of epoch 0, subfigure B is for iteration 800 of epoch 0 and subfigure C is for iteration 3000 of epoch 0.

