# OpenReview forum: "Sparse Attentive Backtracking: Long-Range Credit Assignment in Recurrent Networks"
_ICLR.cc/2018/Conference — Reject_

### Official Review · AnonReviewer3 · 2017-11-28
**SAB Review**

**Rating:** 5
**Confidence:** 4

**Review:**

The paper proposes sparse attentive backtracking, essentially an attention mechanism that performs truncated BPTT around a subset of the selected states.

The early claims regarding biological plausibility seem stretched, at least when applying to this work. The "waiting for life to end to learn" and student study / test analogies were not helpful from an understanding point of view and indeed raised more questions than insight. The latter hippocampal discussion was at least more grounded.

While a strong motivator for this work would be in allowing for higher efficiency on longer BPTT sequences, potentially capturing longer term dependencies, this aspect was not explored to this reviewer's understanding. To ensure clarity, in the character level PTB or Text8 examples, SAB's previous attention was limited to sequences of T = 100 or 180 respectively?
Limiting the truncation to values below the sequence length for the LSTM baselines also appears strange given the standard within the literature is setting sequence length equal to BPTT length. I presume this was done to keep the number of optimizer updates equal?
Another broader question is whether longer term dependencies could be caught at all given the model doesn't feature "exploration" in the reinforcement learning sense, especially for non-trivial longer term dependencies.

When noting the speed of generating a sparsely sourced summary vector (equation 3), it is worth pointing out that weighted summation over vectors in traditional attention is not a limiting factor as it's a very rapid element-wise only operation over already computed states.

For the experiments, I was looking for comparisons to attention over the "LSTM (full BPTT)" window. This experiment would provide an upper bound and an understanding of how much of SAB's improvement may be as a result of simply adding attention to the underlying LSTM models. Even a simpler and fast (cuDNN compatible) attention mechanism such as [a single cuDNN LSTM layer over the input, an attentional mechanism over the results of the first layer (masked to avoid observing timesteps from the future), summed, and then passed into a softmax] would be informative.

Finally, whilst not a deal breaker for introducing new techniques, stronger LSTM baselines help to further underline the efficacy of the technique. For sequential MNIST, a relatively small dataset, previous papers have LSTM models that achieve 98.2% test accuracy (Arjovsky et al, https://arxiv.org/abs/1511.06464) and the IRNN example included as part of the Keras framework achieves 93% out of the box.

Noting similarities to the Transformer architecture and other similar architectures would also be useful. Both are using attention to minimize the length of a gradient's path, though in Transformers it eliminates the RNN entirely. If a Transformer network performed a k=5 convolution or limited RNN run to produce the initial inputs to the Transformer, it would share many similarities to SAB, though without the sparsity.

---

> ### Author Response · Authors · 2018-01-03
> **Response to Reviewer3**
>
> We thank the reviewer for the feedback and comments.
>
> "The early claims regarding biological plausibility seem stretched,..."
>
> Thanks for pointing this out. Our examples did not illustrate the principles well, and we have revised the respective sections to make them more concise.
>
> "While a strong motivator for this work would be in allowing for higher efficiency on longer BPTT sequences, potentially capturing longer term dependencies, ..."
>
> Our experiment on MNIST has a sequence length of 784, which is a good test for long term dependencies. As for the language modeling tasks, it is a common setup to use  T=100 for PTB and T =180 for Text8. We followed the same setup in order to have a comparable baseline to other approaches, such as [1], [2], [3].
> 1. Ha, David, Andrew Dai, and Quoc V. Le. "HyperNetworks." arXiv preprint arXiv:1609.09106 (2016).
> 2. Cooijmans, Tim, et al. "Recurrent batch normalization." arXiv preprint arXiv:1603.09025 (2016).
> 3. Krueger, David, et al. "Zoneout: Regularizing RNNs by randomly preserving hidden activations." arXiv preprint arXiv:1606.01305 (2016).
>
> "Limiting the truncation to values below the sequence length for the LSTM baselines also appears strange..."
>
> Limiting truncation to values below the sequence length is used for Truncated Backpropagation Through Time (TBPTT), which is a technique commonly used to alleviate the computational complexity of longer sequences [4, 5].
> 4. Saon, George, et al. "Unfolded recurrent neural networks for speech recognition." Fifteenth Annual Conference of the International Speech Communication Association. 2014.
> 5. Sak, Haşim, Andrew Senior, and Françoise Beaufays. "Long short-term memory based recurrent neural network architectures for large vocabulary speech recognition." arXiv preprint arXiv:1402.1128 (2014).
>
>
> "Another broader question is whether longer term dependencies could be caught at all ..."
>
> We agree that other mechanisms could be used to foster exploration. In our case, stochastic gradient descent and initial weights which lead to no strong preference do the exploration for us. These methods work well with SAB and they are well-practiced approaches in Deep Learning. Figure 3 in the appendix shows how the attention weight learns to focus on the correct time step as training proceeds.
>
> "For the experiments, I was looking for comparisons to attention over the "LSTM (full BPTT)" window..."
>
> Thanks for pointing this out, this is indeed a nice set of experiments to run. We are currently running those experiments, and will update the paper with the results for LSTM with self-attention.
> Note that LSTM with self-attention requires significantly more GPU memory than SAB, such that the maximum sequence length we can simulate is limited by hardware constraints.
>
> "Finally, whilst not a deal breaker for introducing new techniques, stronger LSTM baselines help to further underline the efficacy of the technique..."
>
> Unfortunately, the experiment was labeled incorrectly. In fact, we run the permuted MNIST experiment, not the sequential MNIST experiment as described. We have fixed this error in the revised version. Our baseline for permuted MNIST is similar to the published baselines.
>
> "Noting similarities to the Transformer architecture and other similar architectures would also be useful..."
>
> There are indeed similarities between the Transformer architecture and SAB. As the reviewer mentions, the Transformer architecture eliminates the RNN entirely. SAB utilizes sparse self-attention to help with RNN training, and hence our motivation is different from the ones in the Transformer network. Although, it is indeed interesting future work to see how the sparsity constraint would work for the Transformer architecture. From what we have seen with our experiments, we strongly suspect that sparsity would not hurt (may be able to help) the Transformer architecture.

---

> ### Author Response · Authors · 2018-01-14
> **Re: SAB Review**
>
> We’d like to thank you again for your review of the paper. We have updated the paper with your suggestions (including  better biological motivation, updated MNIST results, and comparison with self-attention trained using full BPTT).  Would you have any other questions regarding the rebuttal?

---

> ### Author Response · Authors · 2018-01-18
> **Re: SAB Review**
>
> We thank the reviewer again for reviewing our paper. We would like to ask the reviewer if there is any further questions regarding our rebuttal, especially the updated MNIST results and the comparisons with full self-attention.

---

### Official Review · AnonReviewer1 · 2017-11-28
**Sparse attention backtracking, an alternative to (T)BPTT**

**Rating:** 8
**Confidence:** 4

**Review:**

re. Introduction, page 2: Briefly explain here how SAB is different from regular Attention?

Good paper. There's not that much discussion of the proposed SAB compared to regular Attention, perhaps that could be expanded. Also, I suggest summarizing the experimental findings in the Conclusion.

---

> ### Author Response · Authors · 2018-01-03
> **Response to Reviewer 1**
>
> We thank you for your positive review!
>
> Thanks for pointing out the comparison between SAB type attention and regular forms of attention. We are adding a small discussion on the comparison of SAB type attention and regular attention.

---

### Official Review · AnonReviewer4 · 2017-12-07
**SAB combines skip connections with attention**

**Rating:** 5
**Confidence:** 3

**Review:**

This work proposes Sparse Attentive Backtracking, an attention-based approach to incorporating long-range dependencies into RNNs. Through time, a “macrostate” of previous hidden states is accumulated. An attention mechanism is used to select the states within the macro-state most relevant to the current timestep. A weighted combination of these previous states is then added to the hidden state as computed in the ordinary way. This construction allows gradients to flow backwards quickly across longer time scales via the macrostate. The proposed architecture is compared against LSTMs trained with both BPTT and truncated BPTT.

Pros:
- Novel combination of recurrent skip connections with attention.
- The paper is overall written clearly and structured well.


Cons:
- The proposed algorithm is compared against TBPTT but it is unclear the extent to which it is solving the same computational issues TBPTT is designed to solve.
- Design decisions, particularly regarding the attention computation, are not fully explained.

SAB, like TBPTT, allows for more frequent updates to the parameters. However, unlike TBPTT, activations for previous timesteps (even those far in the past) need to be maintained since gradients could flow backwards to them via the macrostate. Thus SAB seems to have higher memory requirements than TBPTT. The empirical results demonstrate that SAB performs slightly better than TBPTT for most tasks in terms of accuracy/CE, but there is no mention of comparing the memory requirements of each. Results demonstrating also whether SAB trains more quickly than the LSTM baselines would be helpful.

The proposed affine form of attention does not appear to actually represent the salience of a microstate and a given time. The second term of the RHS of equation 1 (w_2^T \hat{h}^{(t)}) is canceled out in the subtraction in equation 2, since this term is constant for all i. Thus the attention weights for a given microstate are constant throughout time, which seems undesirable.

The related work discusses skip connections in the context of convolutional nets, but doesn’t mention previous works incorporating skip connections into RNN architectures, such as [1], [2], or [3].

Overall, the combination of recurrent skip connections and attention appears to be novel, but experimental comparisons to other skip connection RNN architectures are missing and thus it is not clear how this work is positioned relative to previous related work.

[1] Lin, Tsungnan, et al. "Learning long-term dependencies in NARX recurrent neural networks." IEEE Transactions on Neural Networks 7.6 (1996): 1329-1338.
[2] Koutnik, Jan, et al. "A clockwork rnn." International Conference on Machine Learning. 2014.
[3] Chang, Shiyu, et al. "Dilated recurrent neural networks." Advances in Neural Information Processing Systems. 2017.

EDIT: I have read the updated paper and the author's rebuttal. I am satisfied with the update to the attention weight formulation. Overall, I still feel that the proposed SAB approach represents a change to the model structure via skip connections. Therefore SAB should also be compared against other approaches that use skip connections, and not just BPTT / TBPTT, which operate on the standard LSTM. Thus to me the experiments are still lacking. However, I think the approach is quite interesting and as such I am revising my rating from 4 to 5.

---

> ### Author Response · Authors · 2018-01-04
> **Response to Reviewer 4 (1/2)**
>
> We thank the reviewer for the feedback and comments.
>
> Q - "Cons:
> - The proposed algorithm is compared against TBPTT but it is unclear the extent to which it is solving the same computational issues TBPTT is designed to solve."
>
> A - Taking “computational issues” to refer to the time- and memory-complexity of the algorithms, we would like to clarify that both time-wise and memory-wise, SAB is more expensive than (T)BPTT. However, unlike full BPTT (which is an inherently sequential algorithm), SAB training is parallelizable given the right hardware (GPU compatibility), which could make SAB as fast as TBPTT. In addition to that, SAB solves an optimization issue: Direct gradient flow from a timestep T_future to dynamically-determined, relevant timesteps T_past potentially arbitrarily far away in the past.
>
> By way of comparison, BPTT does permit gradient flow from any future timestep to any past timestep, but the gradient must flow through T_future - T_past timesteps. In order for any given stream of gradient information to reach arbitrarily far in the past through a finite-capacity channel (Presently, a fixed-size hidden-state vector of 32-bit floating-point numbers), it must compete with and, crucially, survive against other streams all the way along the path backward from T_future to T_past. These other gradient information streams may:
>  -   Be short-range or long-range
>  -   Be fully contained within, partially overlapping with or wholly contain the range [T_past, T_future]
>  -   Concern a greater or lower number of hidden states
>  -   Require more or less precision in each hidden state.
> The survival probability of a gradient information stream therefore decays exponentially with the number of hops it must make in BPTT and the number of competing streams.
>
> TBPTT, due to its truncation of gradient flow, is by design unable to sustain a gradient information stream over a timespan greater than the truncation length. The computational benefit of truncation is parallelizability of the backward pass of the RNN.
>
> Q - "- Design decisions, particularly regarding the attention computation, are not fully explained."
>
> A - Thanks for pointing this out. We agree that the attention mechanism used in the submitted version was not ideal and we have now implemented a slightly different formulation of the sparse attention mechanism, leading to improved results in all tasks. A more detailed description of the problem we have identified and solved, as well as explanations for the design choices, have been added.
>
> Q - "The empirical results demonstrate that SAB performs slightly better than TBPTT for most tasks in terms of accuracy/CE, but there is no mention of comparing the memory requirements of each. "
>
> A - Our empirical results for SAB, full BPTT and truncated BPTT are summarized in Tables 1- 8. Broadly speaking, in the Copying and Permuted MNIST tasks, SAB outperforms full BPTT. For the Adding task, PennTree Bank and Text8 language modeling tasks, SAB  significantly outperforms TBPTT.
>
> - Copying: SAB solves the task for lengths up to 300, and performs much better than full BPTT. For length = 300, SAB accuracy is 98.9% (CE 0.048), whereas full BPTT achieves 35.9% (CE 0.197). Since full BPTT performs much better than TBPTT, SAB significantly outperforms TBPTT of much longer truncation lengths.
>
> - Adding: SAB performs significantly better compared to TBPTT of much longer truncation length. For length = 400, SAB of truncation length = 10 significantly outperforms TBPTT with truncation length = 100.
>
> - PennTree Bank Language Modeling: SAB performs close to full BPTT (BPC of 1.48 vs 1.47; lower is better). SAB (trunc. length = 5) significantly outperforms TBPTT (trunc. length = 20) (BPC 1.48 vs 1.51)
>
> - Text8 Language Modeling: SAB performs close to full BPTT (valid BPC 1.56 vs 1.54), and significantly outperforms TBPTT (BPC 1.56 vc 1.64).
>
> - Permuted MNIST: SAB outperforms full BPTT (accuracy 92.2 vs 91.2). Typically, full BPTT outperforms TBPTT, therefore SAB outperforms TBPTT.
>
> The extra memory required beyond LSTM's basics (for both full BPTT and TBPTT) is the attention mechanism, which is (2h) * (t**2/2k) * (4 bytes), where h is the size of the hidden states and k is the tok k attention selected.

---

> ### Author Response · Authors · 2018-01-04
> **Response to Reviewer 4 (2/2)**
>
> Q - "The proposed affine form of attention does not appear to actually represent the salience of a microstate and a given time. The second term of the RHS of equation 1 (w_2^T \hat{h}^{(t)}) is canceled out in the subtraction in equation 2, since this term is constant for all i. Thus the attention weights for a given microstate are constant throughout time, which seems undesirable."
>
> A - We greatly thank the reviewer for their sharp-eyed identification of this problem. The reviewer is almost entirely correct:
>
> Although the raw attention weights are not constant, the computed sparsified attention weights come dangerously close to being so due to the excessive linearity of the attention and sparsification mechanisms.
>
> The sparsified attention weights are not perfectly constant; However, they will change at most as often as the top-kth selected microstate changes, which happens at most as often as a new microstate gets added to the macro-state, and potentially much more rarely than that.
>
> Moreover, upon further review, we have identified a further linear collapse in the attention mechanism, which caused the attention weights to only be a linear function of the difference between the given microstate and the top-kth microstate.
>
> This is problematic because in principle, if a present hidden state is very similar to a memorized microstate, the attention mechanism should accord it considerable weight, but calculating the attention weight only as a linear function of microstate differences would ignore them by design.
>
> We have modified the attention mechanism so that it is now
> - a concatenation of the hidden and microstate,
> - a linear layer,
> - a hyperbolic tangent non-linearity, and
> - a linear layer again.
> This prevents the linear collapses, and simultaneously gives us both increased accuracies and decreased time to convergence across all tasks. We will update the manuscript with our new results.
>
> Q - "Overall, the combination of recurrent skip connections and attention appears to be novel, but experimental comparisons to other skip connection RNN architectures are missing..."
>
> A - Our work is orthogonal to the work on skip connections in RNNs. SAB is an attention-controlled way of creating a skip connection between two remote points in time in order to avoid the vanishing or exploding gradient issues that plague the learning of long-term dependencies by RNNs. The amount of attention here is controlled by the extent to which an old microstate 'matches' (in some learned way) the current microstate. Skip connections are not new (proposed as early as 1996 with the NARX networks of Lin et al), but using an attention mechanism to select which time steps to pair together and using this to focus the backprop to just a few past time steps is new. It would be interesting future work to see the effect of using different types of skip connections in RNNs.

---

> ### Author Response · Authors · 2018-01-14
> **Re: SAB combines skip connections with attention**
>
> Thank you for your review of the paper and the finding of the cancellation problem in the computation of attention weights! We have eliminated said problem and consequently obtained improved results which significantly outperforms TBPTT and in some cases full BPTT. Lastly, we have noted all of the foregoing in the updated manuscript.
>
> Thanks again for pointing out the shortcomings. Do you have any more questions about the rebuttal, especially as regards the attention mechanism and the strength of the experimental results against TBPTT?

---

### Decision · Program_Chairs · 2018-01-29
**ICLR 2018 Conference Acceptance Decision**

**Decision:**

Reject

**Comment:**

The authors propose to use attention over past time steps to try and solve the gradient flow problem in learning recurrent neural networks. Attention is performed over a subset of past states by a hueristic that boils to selecting best time-steps.

I agree with the authors that they offer a lot of comparisons, but like the reviewers, I am inclined to find the experiments not very convincing of the arguments they are attempting to make.  The model that they propose has similarities to seq2seq in that they use attention to pass more information in the forward pass; in a sense this is a seq2seq model with the same encoder and decoder, and there are parallels to self-attention. The model also has similarities to clockwork RNNs and other skip connection methods.. However, the experiments offered to not tease out these effects. It is unsurprising that a fixed size neural network is unable to do a long copy task perfectly, but an attention model can. What would have been more interesting would have been to explore if other RNN models could have done so. The experiments on pMNIST aren't really compelling as the baselines are far from SOTA (example: https://arxiv.org/pdf/1606.01305.pdf report 0.041 error rate (95.9% test acc) with LSTMs and regularization).  Text8 also shows worse results in full BPTT on LSTM.  If BPTT is consistently better than this method, it defeats the argument that gradient explosion and forgetting over long sequences is a problem for RNNs (one of the motivations offered for this attention model).